# Clinical Contradiction Detection

**Dave Makhervaks**
Technion
Israel Institute of Technology
Haifa, Israel
davem@cs.technion.ac.il

**Plia Gillis**
Tel Aviv University
Tel Aviv, Israel
pliagillis@gmail.com

**Kira Radinsky**
Technion
Israel Institute of Technology
Haifa, Israel
kirar@cs.technion.ac.il

## Abstract

Detecting contradictions in text is essential in determining the validity of the literature and sources that we consume. Medical corpora are riddled with conflicting statements. This is due to the large throughput of new studies and the difficulty in replicating experiments, such as clinical trials. Detecting contradictions in this domain is hard since it requires clinical expertise. We present a distant supervision approach that leverages a medical ontology to build a seed of potential clinical contradictions over 22 million medical abstracts. We automatically build a labeled training dataset consisting of paired clinical sentences that are grounded in an ontology and represent potential medical contradiction. The dataset is used to weakly-supervise state-of-the-art deep learning models showing significant empirical improvements across multiple medical contradiction datasets.

## 1 Introduction

Determining whether a pair of statements is contradictory is foundational to fields including science, politics, and economics. Detecting that statements contradict sheds light on fundamental issues. For instance, mammography is an integral routine in modern cancer risk detection, but there is controversy regarding if certain mammography patterns indicate a cancer risk (Boyd et al., 1984). Recognizing that a certain topic has opposing viewpoints, signifies that the issue may deserve further investigation. Medicine is an interesting domain for contradiction detection, as it is rapidly developing, of high impact, and requires an in-depth understanding of the text. Per the National Library of Medicine, the PubMed (Canese and Weis, 2013) database averaged 900k citations for the years 2018-2021, with a growing trajectory (med, 2022). The publication of contradictory papers is not uncommon in scientific research, as it is part of the process of validating or refuting hypotheses and advancing knowledge. A study on high impact

clinical research found that 16% of established interventions had their outcome refuted (Ioannidis, 2005). Extending this to PubMed, over 5 million articles would disagree with a previous finding.

The problem of contradiction detection has been studied in the task of natural language inference (NLI) on a sentence level. NLI aims to determine whether a pair of sentences are contradictory, entailing, or neutral. Deep learning approaches reach impressive results for this task. Specifically, large language models (LLMs) such as DeBERTaV3 (He et al., 2021) and BioELECTRA (raj Kanakarajan et al., 2021), are considered the state-of-the-art (SOTA) for NLI. However, in medical research, detecting a contradiction is more difficult. Sometimes more context is needed to detect contradiction due to the difficulty of the material. Consider the example below (both interventions are ACE inhibitors):

1. "However, in the valsartan group, significant **improvements in left ventricular hypertrophy and microalbuminuria** were observed."
2. "Although a bedtime dose of doxazosin can significantly **lower the blood pressure**, it can also **increase left ventricular diameter**, thus **increasing the risk of congestive heart failure.**"

Detecting that this pair contradicts requires knowing that *improvements in left ventricular hypertrophy* is a positive outcome, whereas an *increase [in] left ventricular diameter* is a negative outcome in the context of heart failure.

Deep learning methods for NLI require large datasets (Conneau et al., 2017). However, few datasets exist to train such algorithms in the clinical contradiction domain. Time and cost of labeling complex medical corpora, could be a potential reason for this. The MedNLI dataset (Romanov and Shivade, 2018) for instance, required the expert labeling of 4 clinicians over the course of 6 weeks [1]. Yet, MedNLI is fabricated since each of the clin-

---

[1] MIMIC-III certified users can find the MedNLI dataset

icians was given a clinical description of a patient and came up with a contradicting, entailing, and neutral sentence to pair up with that description. In this work, we are interested in naturally occurring sentences in clinical literature as opposed to manually curated texts. We focus on sentences containing clinical outcomes and attempt to identify whether outcomes contradict. We will show that even this step, which does not address the intervention and population, yields an initial system for identifying medical contradictions.

One approach to overcome the lack of large data is distant supervision (Mintz et al., 2009). Distant supervision uses existing knowledge sources to automatically label large amount of data. The labels can be noisy, so the goal is to train robust models that learn meaningful patterns. We propose a novel methodology leveraging distant supervision and a clinical ontology, the Systematized Nomenclature of Medicine Clinical Terms (SNOMED) (Stearns et al., 2001). SNOMED is developed by a large and diverse group of medical experts (Donnelly et al., 2006) and it contains extensive information about clinical terms and their relations. Our method uses knowledge extracted from SNOMED to classify pairs of "naturally occurring", potentially contradictory sentences. PubMed's abstract database is our source for naturally occurring sentences.

We fine-tune SOTA deep learning models on the aforementioned ontology-driven created dataset. We test the results using manually labeled clinical contradiction datasets. The results demonstrate that our distant-supervision-based methodology yields statistically significant improvements of the models for contradiction detection. The average results of 9 different models see an improvement on our main evaluation set (Section 4.1.1) over previous SOTA. Specifically, we find that the improvement is consistent across both small models and those that are considered to be SOTA on NLI tasks, which is the closest task to that of contradiction detection.

The contribution of our work is threefold: (1) We present the novel problem of contradiction analysis of naturally occurring sentences in clinical data. (2) We create a clinical contradiction dataset by using distant supervision over a clinical ontology, yielding improvements of SOTA deep learning models when fine-tuning on it. (3) We empirically evaluate on numerous manually labeled clinical contradiction datasets and show improvements of SOTA

here: https://physionet.org/content/mednli/1.0.0/

models when fine-tuned on the ontology-driven dataset. We make our code publicly available [2].

## 2 Related Work

NLI primarily focuses on textual entailment, starting with the RTE challenges proposed by Dagan et al. (2013) and Dagan et al. (2005). The task involves determining if one sentence can be inferred from another. Over time, data and classification criteria were introduced, including the labeling of contradictions in the third challenge (Giampiccolo et al., 2007). However, the medical domain brings additional challenges requiring clinical expertise.

Despite the complexity of medical literature and the reality of contradictions in publications, there is surprisingly little work in this area. Large NLI corpora contain relatively easy contradiction pairs, partly due to the cost of annotating complex contradictions. The contradiction is often a negation through words like 'not'. An example from a large NLI corpus, MultiNLI (Williams et al., 2017) is:

1. "**Met my** first girlfriend that way."
2. "**I didn't meet my** first girlfriend until later."

Romanov and Shivade (2018) introduce MedNLI, a curated medical NLI dataset, and experiment with baselines, finding InferSent (Conneau et al., 2017) to be the best model. They explore retrofitting (Faruqui et al., 2014) and knowledge-directed attention but observe worse results with retrofitting and minimal improvement with knowledge-directed attention (0.2%-0.3%) compared to their InferSent baseline. In contrast, our approach differs by not using retrofitting or ontology term distances. We also keep the attention mechanism in the models unchanged. Instead, we utilize ontological features specific to clinical findings (Section 3.1) to construct a fine-tuning dataset. We include their methodology as a baseline in Table 3.

Scientific fact-checking is a related task, where a claim is verified against evidence (Wadden et al., 2020; Sarrouti et al., 2021). The work in this field deals with popular claims justified by evidence from a corpus, as opposed to naturally occurring sentences in medical literature. Kotonya and Toni (2020)'s data comes from popular media sources such as the Associated Press and Reuters News.

Alamri and Stevenson (2016) developed a labeled dataset, ManConCorpus, for contradictory cardiovascular research claims in abstracts. This

[2]https://github.com/dmakhervaks/medical-contradictions/tree/main

corpus has pairings between a fabricated query and a naturally occurring claim, thus not dealing with naturally occurring sentences. It is annotated by domain experts. There are works which address contradiction of a clinical query and a claim. Tawfik and Spruit (2018) use hand-crafted features to build a classifier, whereas Yazi et al. (2021) use deep neural networks (see baseline in Table 3). Unlike their approaches, we classify any sentence-pair representing a clinical outcome. To our knowledge, no work evaluates contradiction detection between naturally occurring sentences in clinical literature.

Semantic predications from SemMedDB (Kilicoglu et al., 2012) are commonly used in clinical contradictions. Alamri (2016) uses SemMedDB to construct AutoConCorpus by querying subject-predicate-object tuples. Predicates are 'incompatible' if they belong to different groups. The dataset is in the same query-claim format as ManConCorpus and the 'retrieval' stage is manual. Rosemblat et al. (2019) also use SemMedDB by finding relevant semantic predications with their corresponding opposing predicate pairs. Unlike these approaches, we do not use predicate logic. In addition, our dataset creation is fully automated. We show its efficacy by fine-tuning models on it, which is not done by either of the SemMedDB datasets.

Mintz et al. (2009) introduced distant supervision, which was later extended to knowledge bases like YAGO by Nguyen and Moschitti (2011). Neural networks became popular for distantly-supervised relation extraction (Zeng et al., 2015; Zhang et al., 2019). We apply distant supervision to detect contradictions between sentence pairs representing clinical outcomes. We use weak supervision during fine-tuning with SOTA models and leverage the relational knowledge of a clinical ontology. Unlike common approaches (Smirnova and Cudré-Mauroux, 2018; Purver and Battersby, 2012), we infer contradictions using the structure and attributes of a clinical ontology instead of relying on known relation labels. Our approach yields positive and negative term relations, distinguishing it from traditional distant supervision models (Smirnova and Cudré-Mauroux, 2018). To our knowledge, we are first to use distant supervision for detecting contradictions in the clinical domain.

## 3 Methods

Our method creates a dataset for training or fine-tuning using ontology knowledge, specifically focusing on ontology parts related to clinical findings, which represent the outcomes of clinical observations (Rory, 2023). Unlike mutually exclusive nouns like 'cat' or 'dog' the terms we encounter in this context may have conflicting clinical findings, suggesting a difference in clinical outcome.

### 3.1 SNOMED CT Ontology

SNOMED is an ontology containing over 350,000 clinical terms (Stearns et al., 2001). It has information about a plethora of health concepts, containing attributes such as relationships to other terms and interpretations. The structure of SNOMED allows grouping terms based on their relationships. We hypothesize that this structure coupled with synonyms and antonyms, enables us to create a corpora of potentially contradicting and non-contradicting clinical terms. We use the 2022 edition of SNOMED.

### 3.1.1 SNOMED Node Attributes

Each term in the SNOMED ontology is a node in a tree-like structure designating a clinical finding. These findings have attributes used in determining their relationships. Each node belongs to a group parented by the group root. In addition, they have a simple interpretation which is a defined attribute within the ontology. In Figure 1, the group consists of nodes describing the group root *cardiac output*. The green (right) node has the interpretation - *increased*. Groupings of terms with these attributes have a logical connection, resulting in pairings of contradicting and non-contradicting phrases within the group context. Determining the relationship between a pair of terms is done partially through comparing their interpretations. In Figure 1 the nodes have the interpretations *decreased* and *increased* respectively. We assign the pair an *attribute* label ($A_{i,j}$) of contradiction where $i$ and $j$ are SNOMED terms. In Algorithm 1, $A_{i,j}$ is assigned on Line 12.

Group size can get large. For instance, the group root *Cardiac function* has 275 children. Larger groups may contain child terms that are less closely related compared to smaller groupings. For example *aortic valve regurgitation due to dissection* and *dynamic subaortic stenosis* both pertain to impairments of *cardiac function*, but it would be unfair to consider them related outcomes. Though these large groupings yield many pairings, they may also lead to less accurate ones. Section 5.2 investigates the effects of group sizes.

Below are pairings of contradictions in various medical domains that our methodology yields:

- suppressed urine secretion ↔ polyuria
- elevation of SaO2 ↔ oxygen saturation within reference range
- joint stable ↔ chronic instability of joint

---

**Algorithm 1** SNOMED Traversal

---

1: **function** TRAVERSE($root$)
2:   **for** $n \in root.children$ **do**
3:     **if** $n.num\_childs \leq group\_size$
4:       $pairs \leftarrow$ DET_RELATION($n$)
5:     **end if**
6:   **end for**
7:   **return** $pairs$
8: **end function**

9: **function** DET_RELATION($n$)
10:   $pairs \leftarrow \{\}$
11:   **for** $c_i, c_j \in n.child\_pairs$ **do**
12:     $A_{i,j} \leftarrow$ GET_ATTR_LABEL($c_i, c_j$)
13:     $S_{i,j} \leftarrow$ GET_SYN_LABEL($c_i, c_j$)
14:     $label_{i,j} \leftarrow A_{i,j}$
15:     **if** $S_{i,j}$ = contra **or** $A_{i,j}$ = contra
16:       $label_{i,j} \leftarrow$ contra
17:     **end if**
18:     $pairs \leftarrow pairs \cup \{(label_{i,j}, c_i, c_j)\}$
19:   **end for**
20:   **return** $pairs$
21: **end function**

22: $SNOMED \leftarrow$ TRAVERSE($root$)
23: FINETUNE($Model, SNOMED$)

---

### 3.1.2 Synonyms

After leveraging structure, we examine synonyms and antonyms, which offer a strong signal in an ontology. Grouped clinical terms share a context, thereby allowing the use of simpler indicators to determine their relationship. We word-tokenize each clinical phrase, removing the intersection of the sets of tokens, leaving each with its unique tokens. We assign a SNOMED pair - $i,j$ - a *synonym label* ($S_{i,j}$). A visualization is found in Appendix B.1.

### 3.1.3 Combining Attributes and Synonyms

To combine $A_{i,j}$ and $S_{i,j}$ to form a final $label_{i,j}$, we build a validation set of the publicly available SNOMED term-pairs. Two annotators with domain knowledge label 149 SNOMED phrase-pairs. 70 contradictory and 79 non-contradictory. More details are in Appendix A.1. We find that when $A_{i,j}$ indicates contradiction, then it's likely that $label_{i,j}$

is a contradiction. The same holds for $S_{i,j}$. We define the logic in Lines 15 through 17. We reach 79% accuracy through this heuristic on the human-labeled SNOMED term-pairs with a Cohen's kappa coefficient of 0.853 for inter-annotator agreement.

### 3.2 Ontology-Driven Distant Supervision

Using the relational knowledge extracted from SNOMED, we weakly-supervise naturally occurring sentences in PubMed to build our SNOMED dataset. We fine-tune on this dataset to achieve significant improvements over existing baselines. Algorithm 1 summarizes the procedure. We search PubMed for sentences containing the phrase-pairs discussed in Section 3.1, resulting in a corpus of pairs of sentences. The sentence-pairs are then labeled through distant supervision as explained below. For a given pair of SNOMED terms $(p_1, p_2)$, we label sentences $(s_1, s_2)$ as formalized in Eq.1, where $label \in \{\text{contradiction, non-contradiction}\}$.

$$(p_1 \in s_1) \wedge (p_2 \in s2) \wedge ((p_1, p_2) \in label) \quad (1)$$

This methodology focuses on outcomes, since the SNOMED terms we use are clinical findings. Although there is a chance that there may be differing interventions or participants, we show that even with focusing on outcomes we improve performance on our evaluation sets. Concretely, $p_i$ may be a subset of $s_i$, so there may be information loss (statistics on sizes of $s_i$ are reported in Table 2). Given that the purpose of the SNOMED dataset is to increase the amount of training or fine-tuning data of a model, we find that this introduced noise is acceptable and still yields positive results.

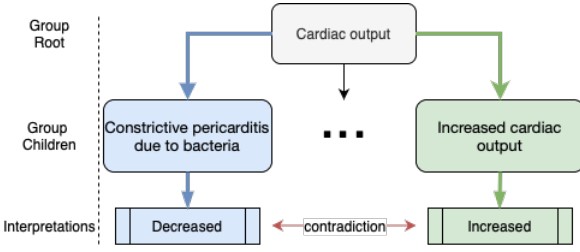

Figure 1: The group with *Cardiac output* as its root. The children depicted have contradicting interpretations.

### 3.3 Filtration

Naively, we pair all sentences satisfying Eq. 1, even if they do not share context. Yet, two sentences containing matching clinical SNOMED terms, may be unrelated. The example below exhibits this:

1. "The present results suggest that the upstream changes in blood flow are transmitted by the velocity **pulse faster** than by the pressure pulse in the microvasculature."
2. "His chest wall was tender and his **pulse slow** but the remainder of his physical examination was normal."

The bolded clinical terms are central to the meaning of the sentences and are contradictory on their own. However, when placed in context they may be less relevant to each other as in the example above. We test stricter criteria for filtering sentence matches, namely using MeSH (Medical Subject Headings) terms (Lipscomb, 2000) and cosine similarity.

MeSH terms categorize articles within PubMed. We hypothesize that sentences drawn from articles with related MeSH terms likely discuss the same topic. Eq. 2 is our formulation for filtering via MeSH terms. $MeSH_i$ and $MeSH_j$ are the sets of MeSH terms for articles containing $sent_i$ and $sent_j$ respectively. Let $t$ be a chosen threshold.

$$\mathbf{1}_A := \begin{cases} 1 & \text{if } \frac{|MeSH_i \cup MeSH_j|}{min(|MeSH_i|, |MeSH_j|)} \geq t\,, \\ 0 & otherwise \end{cases} \quad (2)$$

Using MeSH terms is not perfect. The example below achieves a score of 0.4 per Eq. 2.

1. "In dogs challenged with endotoxin, the inhibition of nitric oxide production **decreased cardiac index** and did not improve survival."
2. "Intra-aortic balloon pumping **increased cardiac index** and aortic distensibility by 24% and 30%, respectively, and reduced myocardial oxygen demand by 31% ($P < .001$ for all alterations)."

Despite overlap in MeSH terms, they are very different - one discusses dogs and the other humans.

The second filtration method measures the cosine similarity between one-hot vectors. Topically related sentences should have a higher one-hot vector cosine similarity. Let $\mathbf{o}_i$ and $\mathbf{o}_j$ be the respective one-hot vectors of $sent_i$ and $sent_j$. Vector lengths are equal to the number of unique words spanning the sentence-pair. We compute the similarities between the vectors as shown in Eq. 3. The dog example above, yields a similarity score of 0.2.

$$\mathbf{1}_A := \begin{cases} 1 & \text{if } cosine(\mathbf{o}_i, \mathbf{o}_j) \geq t\,, \\ 0 & otherwise \end{cases} \quad (3)$$

We experiment with $t$, ultimately choosing $t = 0.35$ based on an external validation set.

Table 1: Cardiology Dataset Breakdown

| Split | Total | Contra | Non-Contra |
|-------|-------|--------|------------|
| Train | 1347 | 571 | 776 |
| Dev | 198 | 100 | 98 |
| Test | 227 | 55 | 172 |

## 4 Empirical Evaluation

### 4.1 Evaluation Datasets

#### 4.1.1 Cardiology Dataset

To assess the quality of the SNOMED dataset, we tweak ManConCorpus. The corpus consists of question-claim pairs. Each question has opposing 'yes', 'no' claims. The claims naturally occur in PubMed and the questions are generated by experts. We convert the corpus by pairing up the sentence claims from PubMed. A pair contradicts if constituent claims answer the question differently. We coin this dataset as *Cardio* (Table 1).

#### 4.1.2 Hard Cardiology Dataset

We observe that models tend to classify sentences as contradictory if negations appear. For example:

1. "Our results indicate that atorvastatin therapy significantly improves BP control in hyperlipidemic hypertensive patients."
2. "Administration of a statin in hypertensive patients in whom blood pressure is effectively reduced by concomitant antihypertensive treatment **does not have** an additional blood pressure lowering effect."

We construct a version of Cardio through removing negation words. This version removes model reliance on such words to make a correct prediction. We coin this dataset as *Hard-Cardio*.

#### 4.1.3 MedNLI Datasets

Inspired by SNLI (Bowman et al., 2015), MedNLI was created with a focus on the clinical domain (Romanov and Shivade, 2018). It was curated for 6 weeks, borrowing the time of 4 doctors. MedNLI consists of sentence-pairs which are grouped into triples - a contradictory, entailing, and neutral pair. The sentences are not naturally occurring in existing medical literature. The premise is shared across the three pairs, but each have a different hypothesis, yielding a different label. Since MedNLI deals with a 3-class problem, we relabel the dataset by making {*entailment, neutral*} map to *non-contradiction*.

Our focus is to show that the SNOMED dataset is as powerful as the MedNLI dataset, without the

need for manual intervention. The baseline on the relabeled version of MedNLI gives high results (Appendix G), so adding additional data makes little change. The largest labeled datasets containing naturally occurring sentences are at most hundreds of sentences. Thus, we sample 100 instances from MedNLI's train-split and report results on that.

To explore fields outside of cardiology, we create versions of MedNLI focused on gynecology (GYN), endocrinology (ENDO), obstetrics (OB), and Surgery. The annotators introduced in Section 3.1.3 filter the data. We sample from the train-split in the same fashion as explained above. These datasets also have a 2-class label structure as explained in Section 4.1.3 (Appendix A.2 for details).

## 4.2 SNOMED Dataset Analysis

Table 2 presents statistics of the SNOMED dataset used for weak supervision, including the number of analyzed articles and the sentence counts in PubMed containing SNOMED ontology terms. See Appendix D for qualitative examples of our dataset.

### 4.2.1 Phrase Matching Noise

The proposed phrase-matching introduces noise when $p_i$ is not central to the meaning of $s_i$. To approximate this noise, we sample 100 sentences from the SNOMED dataset. A human annotator was asked to evaluate if $p_i$ contributes to the central message of $s_i$. We observe 91% accuracy.

### 4.2.2 SNOMED Labeling Noise

The SNOMED dataset may contain noise. We sample 100 instances from it and manually label the sentence-pairs, similar to Mintz et al. (2009). The annotators label each pair as containing contradictory or non-contradictory elements. The gold label is compared to the distantly-supervised label. We extract positive and negative relations from our ontology, thus reporting accuracy to indicate the method's effectiveness. We achieve 82% accuracy, surpassing previous noise analyses (Mintz et al., 2009), possibly due to rich ontology information.

## 4.3 Baseline Models

Romanov and Shivade (2018) use InferSent with knowledge-directed attention to achieve their top results on MedNLI. Yazi et al. (2021) achieve the SOTA on the ManConCorpus, which we turn into the Cardio corpus (Section 4.1.1). They concatenate BERT embeddings for their question and claim, feeding the input into a Siamese-like feed

Table 2: SNOMED Dataset

| Sentence length: | |
|---|---|
| NLTK token count | 25.1 |
| BioGPT token count | 29.4 |
| BioELECTRA token count | 30.7 |
| BERT-Base token count | 36.8 |
| **Total Dataset Statistics:** | |
| SNOMED term matches in PubMed | 4.99M |
| Number of articles | 2.87M |
| Number of qualifying pairs in SNOMED | 0.63M |

forward network. We use the same hyperparameters as the authors to replicate their results.

We evaluate an additional 9 LLMs and compare their performance when they are fine-tuned on the SNOMED dataset versus without. The task of classifying contradiction is a subtask of NLI, thus we use models that top leaderboards for the MNLI and MedNLI datasets - namely DeBERTaV3-Base, ALBERT (Lan et al., 2019), and BioELECTRA. ELECTRA (Clark et al., 2020) and BERT (Devlin et al., 2018) are also included as they are performant architectures. Additionally, success on small models is important, as they require less computing resources and may allow the SNOMED dataset to have a stronger influence during fine-tuning. Thus, we include BERT-Small (Turc et al., 2019), ELECTRA-Small, and DeBERTaV3-Small (He et al., 2021). Finally, we include BioGPT (Luo et al., 2022) for completeness, as it has a decoder and is also pre-trained on biological data. Table 3 breaks down the number of parameters per model.

The LLMs have the same high-level architecture, so we use the functionalities of HuggingFace (Wolf et al., 2019) and the Sentence-Transformer library (Reimers and Gurevych, 2019). We pass sentence-pairs as input into the network and add a binary classification head to the model body. Hyperparameters come from the Sentence-Transformer library, except for training batch size - 8 for models above 30M parameters and 16 for models under 30M parameters. Each model is tuned with the SNOMED dataset. It uses a group size of 25, sampling 10 sentence-pairs from PubMed for every SNOMED term-pair. These hyperparameters are determined through ablation tests on the Cardio validation set.

## 5 Empirical Results

### 5.1 Main Result

Table 3 summarizes our findings. *Base* denotes fine-tuning a model *only* on the training split of the corresponding evaluation dataset. *Ours* de-

Table 3: Performance of Models tuned with SNOMED vs. Without

| Dataset | Method | Algorithm (Number of Params) | | | | | | | | | | |
|---|---|---|---|---|---|---|---|---|---|---|---|---|
| | | ALBERT Base (11.7M) | ELECTRA Small (13.5M) | BERT Small (28.8M) | ELECTRA Base (109.5M) | BERT Base (109.5M) | Bio-ELECTRA (109.5M) | DeBERTa Small (141.9M) | DeBERTa Base (184.4M) | Bio—GPT (346.8M) | (Yazi et al., 2021) | (Romanov and Shivade, 2018) |
| Cardio | Base | 0.911 | 0.877 | 0.858 | 0.863 | **0.914** | 0.880 | 0.885 | 0.861 | 0.858 | 0.858 | 0.824 |
| | Ours | **0.928** | **0.947*** | **0.958*** | **0.892** | 0.878 | **0.925** | **0.931*** | **0.942*** | **0.930*** | - | - |
| Hard-Cardio | Base | 0.876 | 0.785 | 0.717 | 0.847 | **0.803** | 0.850 | 0.842 | 0.845 | 0.762 | 0.687 | 0.688 |
| | Ours | **0.925*** | **0.853*** | **0.794*** | **0.873** | 0.791 | **0.925*** | **0.917*** | **0.936*** | **0.871*** | - | - |
| MedNLI-General | Base | 0.541 | 0.516 | 0.492 | 0.537 | 0.661 | 0.539 | 0.551 | 0.565 | 0.725 | 0.529 | 0.643 |
| | Ours | **0.682*** | **0.636*** | **0.605*** | **0.754*** | **0.759*** | **0.779*** | **0.695*** | **0.774*** | **0.800*** | - | - |
| MedNLI-Cardio | Base | 0.749 | 0.542 | 0.553 | 0.600 | 0.759 | 0.597 | 0.589 | 0.672 | **0.840** | 0.557 | 0.738 |
| | Ours | **0.808** | **0.648*** | **0.692*** | **0.785*** | **0.794** | **0.834*** | **0.777*** | **0.864*** | 0.833 | - | - |
| MedNLI-GYN | Base | 0.492 | 0.533 | 0.600 | 0.525 | 0.575 | 0.558 | 0.592 | 0.625 | 0.583 | 0.508 | 0.708 |
| | Ours | **0.608** | **0.617** | **0.767** | **0.758** | **0.792** | **0.808** | **0.683** | **0.825** | **0.783** | - | - |
| MedNLI-ENDO | Base | 0.698 | 0.525 | 0.567 | 0.584 | 0.639 | 0.601 | 0.522 | 0.601 | 0.840 | 0.560 | 0.707 |
| | Ours | **0.860*** | **0.690** | **0.725** | **0.793*** | **0.867*** | **0.860*** | **0.852*** | **0.883*** | **0.878** | - | - |
| MedNLI-OB | Base | 0.532 | 0.502 | 0.513 | 0.505 | 0.557 | 0.549 | 0.502 | 0.579 | 0.507 | 0.505 | 0.538 |
| | Ours | **0.616** | **0.542** | **0.581** | **0.667** | **0.625** | **0.702*** | **0.618** | **0.740*** | **0.630** | - | - |
| MedNLI-Surgery | Base | 0.708 | 0.502 | 0.555 | 0.681 | 0.842 | 0.669 | 0.576 | 0.691 | 0.925 | 0.602 | 0.898 |
| | Ours | **0.892*** | **0.668** | **0.807*** | **0.912*** | **0.903** | **0.925*** | **0.808*** | **0.884*** | **0.940** | - | - |

notes fine-tuning on both the SNOMED dataset and the training split of the corresponding evaluation dataset. Both *Base* and *Ours* are evaluated on the same test sets. We measure the area under the ROC curve of each baseline and verify statistical significance through Delong's test (DeLong et al., 1988). Significant differences are marked with an asterisk (*). Across most dataset-model combinations, the weak supervision of the SNOMED dataset reaches superior results compared to fine-tuning only on the original dataset. The result holds for Romanov and Shivade (2018)'s InferSent model and Yazi et al. (2021)'s SOTA model (Appendix F).

Cardio and Hard-Cardio are difficult datasets of potentially contradicting, naturally occurring pairs in PubMed. The performance on Hard-Cardio drops relative to Cardio as expected. This verifies our hypothesis that removing negations makes the problem more difficult. Fine-tuning on the SNOMED dataset improves the baselines for 8 out of 9 models we evaluate for both datasets.

On synthetically created common datasets, such as MedNLI, our methodology improves over *all* baselines for this corpus. Improvements are also consistent for sub-specialties when fine-tuning on SNOMED. This shows the scalability of our methods for clinical contradiction detection through different fields within healthcare.

Further, we see a trend where small models may be more affected by fine-tuning on SNOMED. All of the evaluation datasets improve over the baseline on *every* model under 30 million parameters.

## 5.2 Ablation Studies

### 5.2.1 Group and Sentence Samples Size

We explain SNOMED term grouping in Section 3.1 and illustrate in Figure 1. Group size and pairing quality may be closely related. Larger groupings tend to have more terms which are less related to each other as explained in Section 3.1.1. Thus, we experiment with SNOMED datasets based on terms belonging to groups of at most 6, 12, 25, and 50.

During dataset creation, we choose how many sentence-pairs to sample per SNOMED pairing. In Figure 2, each line with a different color/marker represents a different number of samples averaged across the models. The ablations we perform include 10, 25, and 50 samples per pairing.

Figure 2 shows 10 samples outperforms higher sampling numbers for most group sizes. Increased sampling results in over-saturation of certain term-pairs. The best group size is 25 for small models and 12 for large models. These numbers strike the balance of a large amount of SNOMED phrase-pairings and accurate relationships (as discussed in Section 3.1.1). Smaller models may benefit more from larger group sizes, because they have a more limited base knowledge than those of large models.

### 5.2.2 Filtering Based on Similarity

To enhance the relevance of sampled sentences, we explore the use of high MeSH term or cosine similarity for filtering (Section 3.2). Figure 3 shows the relation between the filtration methods. Continuing the ablation analysis from Figure 2, we set the num-

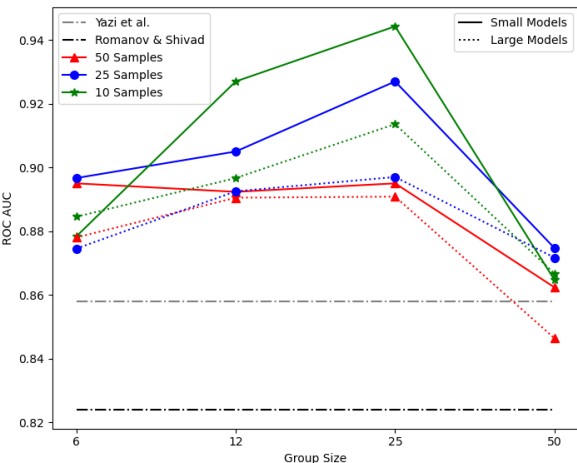

Figure 2: Small and large model performance across group sizes and sample numbers. Reported on Cardio.

ber of samples at 10 and the group size at 25. The cosine similarity outperforms both no-filtering and MeSH. Although MeSH terms are useful, they are tagged on the article-level, thereby not providing the same topic granularity as one-hot vectors.

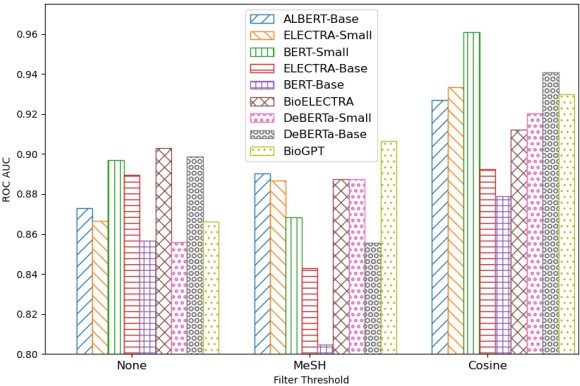

Figure 3: Performance across filtration methods. Number of samples is 10 and group size is 25. Reported on Cardio.

## 5.3 Qualitative Application on Real Abstracts

In an extrinsic experiment, we aim to show the value of our method in a more realistic clinical setting. Thus far, we evaluated our methodology on the sentence level. We now present a preliminary system to see if our method can extract contradictory abstracts. Results are evaluated by clinicians.

### 5.3.1 Representative Sentences

Our models are trained on sentence-level data, so we find a sentence representative of the abstract. This is done automatically by finding the sentence with the largest cosine similarity to the title.

### 5.3.2 Similar Articles

To find similar articles, we use the PubMed Related Articles metric (Lin and Wilbur, 2007). Castro et al. (2015) find that this is an effective metric for semantic simalrity between articles when compared to others like BM25 (Robertson et al., 1980).

### 5.3.3 Finding Contradictions

We sample 850 abstracts. 51 are flagged as potentially contradictory by a BERT-Small model finetuned on the SNOMED dataset. After filtering sentences without annotator-identified outcomes, we have 24 remaining. Among them, 9 are labeled as contradictory by our annotators. 12 false positives are attributed to mismatch in intervention or experimental design, despite contradictory outcomes. For example if the abstract discussed different drug types, but the sentences had opposite findings of a patient's blood pressure. 3 were not contradictory. In future work we intend to improve our methodology by also focusing on the intervention component and making this application more robust. We attach examples of contradictory abstracts in Appendix E.

## 6 Conclusions

Contradiction detection is central to many fields, but is especially important in medicine due to human impact and rapid clinical research growth. Though contradictions are a subfield of NLI, there is less work in the clinical domain. Medical contradictions require context and domain knowledge, making them complex. Labeling datasets which produce effective deep learning models is costly.

We introduce a novel method of using a clinical ontology to weakly-supervise the creation of a potentially contradicting dataset with naturally occurring sentences. We coin it the SNOMED dataset. Empirical results suggest that fine-tuning on the SNOMED dataset results in consistent improvement across SOTA models over diverse evaluation datasets spanning multiple medical specialties. We find a balance between term group size and the number of PubMed sentence samples per pairing. Additionally, better results are achieved by filtering the PubMed sentences included in our dataset.

For future exploration we suggest investigating more robust sentence filtration methods, such as topic modeling or sentence embedding similarity. Looking into how other clinical ontologies can be paired with SNOMED may also be fruitful.

## Limitations

We believe in the novelty of our work and the impact it may provide, but there are some limitations. This methodology is limited to SNOMED terms, many of which do not appear within PubMed. Due to the evolving nature of knowledge bases, terminology and information changes, potentially altering relations between terms. In addition, the structure we extract from the clinical ontology is not ground-truth, yielding noise during dataset creation as discussed in Section 4.2.

In the creation of the SNOMED dataset we focus on contradicting outcomes. Incorporating intervention and participant understanding into our methodology is important and we leave this for future work. The final qualitative application on real abstracts (Section 5.3) is a preliminary system which can be improved in finding the most representative sentence of an abstract and only including those sentences with outcomes reflecting the main finding. Our system is also limited to single sentences which is a constraint, especially when wanting to compare full abstracts.

## Ethical Considerations

Whenever working within the clinical domain, ethical considerations are crucial. The data that we work with is all rooted in already publicly available corpora and PubMed. To the best of our knowledge the data we use does not contain any personal information of any humans involved in clinical trials. There is a potential risk of over representing common diseases and outcomes in our dataset, thereby not including enough data about other outcomes. If our approach were to be implemented in a real medical environment, there could also be repercussions. Due to the imperfect nature the methodology, any label predicted by our model would need be used as a guide for leading the researcher or doctor, as opposed to the ground truth. An example scenario may be choosing to use drug $A$ to treat disease $B$ since our model does not find any contradictory evidence regarding the effectiveness of drug $A$. However, the decision of the physician should take into account that the model may not always be accurate.

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

## A Annotation

As mentioned in Section 3.1.3, we work with annotators with domain knowledge. The annotators were used due to their expertise in the field.

### A.1 SNOMED Term-Pairs

The annotators labeled 149 SNOMED term-pairs as either contradictory or non-contradictory. They were provided with a list of pairs, without any additional information about the ontological structure they came from. This was done in order to preserve fairness and integrity during the labeling process. The instructions were to come up with a binary label for each of the pairs.

### A.2 Filtering MedNLI

A human annotator also helped with coming up with a list of sub-words which served as indicators for particular fields of medicine. For example, the sub-words *vulv* and *gyno*, are indicative of gynecology. These word lists were used to create the variations of MedNLI discussed in Section 4.1.3. You can find the lists of words in the code that is released with the paper.

## B Additional Methodology Details

### B.1 Synonym Extraction

Synonym extraction is a part of our methodology which is explained in Section 3.1.2. Figure 4 provides a depiction of this for the clinical terms *shortened p wave* and *prolonged p wave*. The respective unique tokens are *shortened* and *prolonged*. Since the unique tokens are antonyms, the *synonym* label for the pair is a contradiction. In Algorithm 1, the *synonym* label ($S_{i,j}$) is assigned on Line 13. Similarly, if the respective tokens are synonyms, then $S_{i,j}$ would be a non-contradiction.

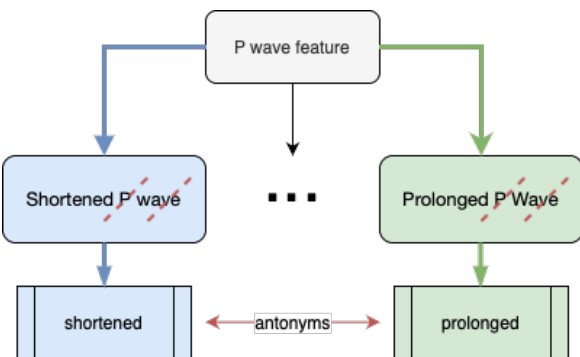

Figure 4: The terms *shortened p wave* and *prolonged p wave* are simplified to just *shortened* and *prolonged* after their common words are removed. The remaining words are antonyms.

## C Additional Dataset Details

We include some additional details to the breakdown of the evaluation datasets. In particular, regarding the token lengths of the datasets. In Table 2 we include the sentence length breakdown according to various tokenizers of the SNOMED dataset. In the appendix we also include the breakdown of the Cardio dataset as well as the MedNLI dataset (Tables 4 and 5 respectively). Although our SNOMED dataset and Cardio dataset contain roughly the same number of tokens per sentence, the MedNLI dataset contains roughly half the number of tokens per sentence. This may serve as an indicator to the decreased difficulty of MedNLI as well as evidence that that sentences are not naturally occurring.

## D SNOMED Dataset Examples

We include several randomly sampled examples from the SNOMED dataset. The data is also publicly available.

Table 4: Cardio Dataset Additional Details

| Sentence length: | |
| --- | --- |
| NLTK token count | 26.7 |
| BioGPT token count | 30.6 |
| BioELECTRA token count | 31.8 |
| BERT-Base token count | 37.2 |

Table 5: MedNLI Additional Details

| Sentence length: | |
| --- | --- |
| NLTK token count | 13.2 |
| BioGPT token count | 16.3 |
| BioELECTRA token count | 17.1 |
| BERT-Base token count | 19.1 |

## D.1  Contradiction Examples

**Example:**

Sentence 1: the plasma cck and luminal content of lcrf were measured by specific radioimmunoassays.;bile-pancreatic juice diversion significantly increased pancreatic secretion plasma cck and lcrf levels.

Sentence 2: blockade of the cck receptor results in decreased pancreatic secretion and atrophy.

**Example:**

Sentence 1: although the mutant does not swim still it is able to move and perform photobehavior.

Sentence 2: whereas the chev mutants still produced both types of flagella and were able to swim and swarm.

## D.2  Non-Contradiction Examples

**Example**

Sentence 1: computed tomography (ct) scans showed bilateral contracted kidneys with a mass projecting from the lower pole of the right kidney.

Sentence 2: ultrasonography and computed tomography revealed a masslike expansion involving the upper pole of an otherwise small right kidney.

**Example**

Sentence 1: hearing loss tinnitus hyperacusis and difficulty hearing in noise remain persistent and in some cases progressive complaints for patients.

Sentence 2: chief complaints were longstanding localized pain and hearing difficulty.

## E  Wild Contradictory Abstracts

The numbers are the PMID of the articles. More examples can be found in the data directory attached with the paper submission. **Example**

Abstract 27045229: This meta-analysis investigated the effects of preoperative prophylactic intraaortic balloon pump placement on postoperative renal function and short-term death of high-risk patients undergoing coronary artery bypass grafting. We found that preoperative prophylactic intraaortic balloon pump support reduced the incidence of coronary artery bypass grafting-associated acute kidney injury and short-term death and dramatically decreased the incidence of postoperative renal replacement therapy by 82% compared with high-risk patients without the procedure. This is the first meta-analysis to demonstrate significant beneficial effects of preoperative prophylactic intraaortic balloon pump on renal function in high-risk patients undergoing coronary artery bypass grafting.

Abstract 29863415: Background Urgent or emergency coronary artery bypass grafting in patients with acute coronary syndrome is associated with increased morbidity and mortality. We investigated the effects of preoperative intraaortic balloon pump support in this high-risk patient cohort. Methods Our institutional database was retrospectively reviewed for patients with acute coronary syndrome and an urgent or emergency indication for coronary artery bypass from April 2010 to December 2016. Data of 1066 patients were analyzed. We assessed the impact of preoperative intraaortic balloon pump therapy on postoperative mortality and major adverse cardiovascular and cerebrovascular

events, and performed propensity-score matching. Results Intraaortic balloon pump support was implemented in 223 (20.9%) patients: 55 (5.2%) preoperatively and 168 (15.8%) intra- or postoperatively. Overall hospital mortality was 8.8%. Patients with a preoperative intraaortic balloon pump had increased mortality (11/55, 20%) compared to controls ( p = 0.006). After propensity-score matching, all-cause mortality (20.0% vs. 18.2%, p = 0.834), cardiac mortality (18.2% vs. 14.5%, p = 0.651), and major adverse cardiovascular and cerebrovascular events (29.1% vs. 27.3%, p = 0.855) were comparable between groups. Conclusions Preoperative intraaortic balloon pump support does not confer any additional clinical benefit on patients undergoing coronary artery bypass grafting for acute coronary syndrome.

## Example

Abstract 8576790: GENETIC PREDISPOSITION: Insulin resistance and reactive hyperinsulinemia occur not only with obesity, impaired glucose tolerance or non-insulin-dependent (type 2) diabetes mellitus, but also in many non-obese, non-diabetic patients with essential hypertension and their currently normotensive, lean, young offspring, as well as in some other conditions known to promote hypertension. Insulin resistance impairs glucose tolerance, while insulin resistance and/or hyperinsulinemia promote dyslipidemia, body fat deposition and probably atherogenesis. Therefore, the common coexistence of a genetic predisposition for hypertension with insulin resistance helps to explain the frequent, although temporally often dissociated, occurrence of hypertension together with dyslipidemia, obesity and type 2 diabetes in a given patient. INSULIN RESISTANCE AND HYPERINSULINEMIA AS SLOW PRESSOR MECHANISMS: In the pathogenesis of hypertension, inappropriate vasoconstriction (due to an imbalance of vasoactive substances and/or raised cytosolic calcium) and/or structural vasculopathy is particularly impor-

tant. Among the mosaic of assumed pressor mechanisms, distinct Na+ retention is almost invariably involved in diabetes mellitus, while sympathetic activation tends to occur in essential hypertension, particularly in association with obesity. Insulin resistance may develop as a consequence of an intracellular excess of Ca2+ or a decrease in Mg2+, an impaired insulin-mediated rise in skeletal muscle blood flow, increased sympathetic activity or excess body weight. Acute hyperinsulinemia causes arterial vasodilation on one hand and increases sympathetic activity and renal Na+ reabsorption on the other. Chronically, hyperinsulinemia may promote cardiovascular muscle cell proliferation and atherogenesis, while insulin resistance may be associated with certain transmembraneous cation transporters, leading to an increase in cytosolic Ca2+. Hyperinsulinemia and/or insulin resistance may also be associated with an increased blood pressure sensitivity to high salt intake. In the mosaic of many different blood pressure-raising mechanisms, insulin resistance and/or hyperinsulinemia is likely to represent an amplifying slow or very slow pressor factor.

Abstract 7700881: Resistance to the metabolic effects of insulin and compensatory hyperinsulinemia have been postulated to mediate human essential hypertension, especially when associated with obesity. Evidence supporting this hypothesis has come mainly from epidemiological studies showing correlations between insulin resistance, hyperinsulinemia, and blood pressure, and from short-term studies suggesting that insulin has renal and sympathetic effects that could raise blood pressure if the effects were sustained. However, there have been no studies demonstrating a direct causal relationship between chronic hypertension and insulin resistance or hyperinsulinemia in humans. The few long-term studies that have been conducted in dogs and humans do not support the hypothesis that hyperinsulinemia causes

hypertension or potentiates the hypertensive effects of other pressor agents such as angiotensin II or increased adrenergic tone. To the contrary, multiple studies in dogs and in humans suggest that the vasodilator action of insulin tends to reduce blood pressure. Although resistance to insulin's metabolic effects has been suggested to be essential for hyperinsulinemia to cause hypertension, chronic increases in plasma insulin concentrations do not cause hypertension in dogs or humans even in the presence of insulin resistance. Also, recent studies have also shown that the blood pressure-lowering effects of antihyperglycemic agents, initially believed to lower blood pressure by decreasing insulin resistance, may be unrelated to their effects on insulin sensitivity. Obesity appears to be a key factor in accounting for correlations between insulin resistance, hyperinsulinemia, and hypertension, but increased blood pressure in obesity does not appear to be mediated by insulin resistance and hyperinsulinemia. Although insulin resistance and hyperinsulinemia may not be directly linked to hypertension, there is increasing evidence that metabolic abnormalities associated with insulin resistance may increase the risk of cardiovascular disease (e.g., coronary artery disease) associated with hypertension and Type II diabetes. For this reason, further studies of the long-term effects of insulin resistance on cardiovascular, renal, and metabolic functions are needed.

**Example**

Abstract 25973956: Colorectal cancer (CRC) can be classified into different types. Chromosomal instable (CIN) colon cancers are thought to be the most common type of colon cancer. The risk of developing a CIN-related CRC is due in part to inherited risk factors. Genome-wide association studies have yielded over 40 single nucleotide polymorphisms (SNPs) associated with CRC risk, but these only account for a subset of risk alleles. Some of this missing heritability may be due to gene-gene interactions.

We developed a strategy to identify interacting candidate genes/loci for CRC risk that utilizes both linkage and RNA-seq data from mouse models in combination with allele-specific imbalance (ASI) studies in human tumors. We applied our strategy to three previously identified CRC susceptibility loci in the mouse that show evidence of genetic interaction: Scc4, Scc5 and Scc13. 525 SNPs from genes showing differential expression in the mouse and/or a previous role in cancer from the literature were evaluated for allele-specific imbalance in 194 paired human normal/tumor DNAs from CIN-related CRCs. One hundred three SNPs showing suggestive evidence of ASI (31 variants with uncorrected p values ¡ 0.05) were genotyped in a validation set of 296 paired DNAs. Two variants in SNX10 (SCC13) showed significant evidence of allelic selection after multiple comparisons testing. Future studies will evaluate the role of these variants in combination with interacting genetic partners in colon cancer risk in mouse and humans.

Abstract 21314996: Common single-nucleotide polymorphisms (SNPs) in ten chromosomal loci have been shown to predispose to colorectal cancer (CRC) in genome-wide association studies. A plausible biological mechanism of CRC susceptibility associated with genetic variation has so far only been proposed for three loci, each pointing to variants that affect gene expression through distant regulatory elements. In this study, we aimed to gain insight into the molecular basis of seven low-penetrance CRC loci tagged by rs4779584 at 15q13, rs10795668 at 10p14, rs3802842 at 11q23, rs4444235 at 14q22, rs9929218 at 16q22, rs10411210 at 19q13, and rs961253 at 20p12.

Possible somatic gain of the risk allele or loss of the protective allele was studied by analyzing allelic imbalance in tumour and corresponding normal tissue samples of heterozygous patients. Functional variants were searched from in silico predicted enhancer elements locat-

ing inside the CRC-associating linkage-disequilibrium regions.

No allelic imbalance targeting the SNPs was observed at any of the seven loci. Altogether, 12 SNPs that were predicted to disrupt potential transcription factor binding sequences were genotyped in the same population-based case-control series as the seven tagging SNPs originally. None showed association with CRC.

The results of the allelic imbalance analysis suggest that the seven CRC risk variants are not somatically selected for in the neoplastic progression. The bioinformatic approach was unable to pinpoint cancer-causing variants at any of the seven loci. While it is possible that many of the predisposition loci for CRC are involved in control of gene expression by targeting transcription factor binding sites, also other possibilities, such as regulatory RNAs, should be considered.

## F  SNOMED Tuning on Baselines

We decide to also fine-tune the baselines of Romanov and Shivade (2018) and Yazi et al. (2021) on the SNOMED dataset. Table 6 summarizes these results. Almost *all* dataset-model combinations improve with the addition of the SNOMED dataset. This result is consistent with our main results.

## G  Full MedNLI Dataset

For completeness, we also report results on the full MedNLI dataset. Results can be see in Table 7. Notably, there are no statistically significant results. Although fine-tuning with the SNOMED dataset yields better results in majority of the models, we see that results are not statistically significant in any of the cases. Therefore, we hypothesize that there is over-saturation which occurs at this stage.

Table 6: Performance of Models tuned with SNOMED vs. Without

|  |  | Algorithm | |
|---|---|---|---|
| Dataset | Method | (Yazi et al., 2021) | (Romanov and Shivade, 2018) |
| Cardio | Base | 0.858 | 0.824 |
|  | Ours | **0.917*** | **0.881*** |
| Hard-Cardio | Base | 0.687 | 0.688 |
|  | Ours | **0.717** | **0.728** |
| MedNLI-General | Base | 0.529 | 0.643 |
|  | Ours | **0.643*** | **0.659** |
| MedNLI-Cardio | Base | 0.557 | 0.738 |
|  | Ours | **0.686*** | 0.738 |
| MedNLI-GYN | Base | 0.508 | 0.708 |
|  | Ours | **0.617** | **0.775** |
| MedNLI-ENDO | Base | 0.560 | 0.707 |
|  | Ours | **0.771** | **0.748** |
| MedNLI-OB | Base | 0.505 | 0.538 |
|  | Ours | **0.595** | **0.605** |
| MedNLI-Surgery | Base | 0.602 | **0.898** |
|  | Ours | **0.895*** | 0.855 |

Table 7: Performance of Models tuned with SNOMED vs. Without

| Dataset | Method | Algorithm (Number of Params) | | | | | | | | | | |
|---|---|---|---|---|---|---|---|---|---|---|---|---|
|  |  | ALBERT Base (11.7M) | ELECTRA Small (13.5M) | BERT Small (28.8M) | ELECTRA Base (109.5M) | BERT Base (109.5M) | Bio-ELECTRA (109.5M) | DeBERTa Small (141.9M) | DeBERTa Base (184.4M) | Bio—GPT (346.8M) | (Yazi et al., 2021) | (Romanov and Shivade, 2018) |
| MedNLI | Base | 0.946 | 0.934 | **0.936** | 0.962 | 0.951 | 0.973 | **0.968** | **0.977** | **0.962** | 0.934 | **0.930** |
|  | Ours | **0.951** | **0.934** | 0.933 | **0.962** | **0.952** | **0.973** | 0.966 | 0.971 | 0.961 | **0.938** | 0.923 |