# OpenReview forum: "Clinical Contradiction Detection"
_EMNLP/2023/Conference — EMNLP 2023 Main_

### Official Review · Reviewer_G3pb · 2023-07-23

**Soundness:** 3

**Excitement:**

3: Ambivalent: It has merits (e.g., it reports state-of-the-art results, the idea is nice), but there are key weaknesses (e.g., it describes incremental work), and it can significantly benefit from another round of revision. However, I won't object to accepting it if my co-reviewers champion it.

**Paper Topic And Main Contributions:**

The study introduced a weakly-supervised approach by using the SNOMED-CT ontology to supplement the training data in developing deep learning classifiers that identify contradicting medical sentences. An antonym lexicon was created by leveraging the opposite SNOMED-CT interpretation values associated with each clinical finding concept and the concept terms under specified subtrees. The lexicon was then used with compositional heuristics to assign pairs of PubMed abstract sentences as contradictory or non-contradictory, which were used to train deep learning natural language inference (NLI) models and evaluated on several existing medical NLI datasets. Major improvements were observed with using these ontology-supplemented training data. Choosing proper subtree sampling scope and pre-denoising irrelevant sentence pairs were also found beneficial.

**Questions For The Authors:**

Question A: Is the label at Algorithm 1 Line 13 manually assigned? If automated, how?
Question B: Is Mintz et al’s 2009 work a valid comparison in the context of section 4.2.2? Please justify.
Question C: Are you able to publicly share the raw annotations based on which the 91% and 82% accuracies are claimed in 4.2.1 and 4.2.2?
Question D: Were the section 3.1.3’s 149 manually verified pairs the only ones used in training the NLI models? What is the difference between the 79% accuracy in 3.1.3 versus the 82% accuracy in in 4.2.2?

**Reasons To Accept:**

I think this is a well-chosen, meaningful challenge in terms of matching what the NLI technical framework could be conveniently adapted to tackle in the medical domain. Despite the underdeveloped methods, the genuine thought process and ambition to solve a hard problem are appreciated.

**Reasons To Reject:**

Many of the proposed heuristics are crude (e.g., the example at line 940 does not clearly differentiate the subject “the mutant” vs “the chev mutants”), which cast doubt on the promising results in Table 3 with regard to the validity of the sentence-based evaluation. Determining contradictions in medical research is very context-sensitive, and even minor misalignment of the settings can make a detected pair useless. A more realistic test scenario is probably what is reported in section 5.3.3, with a precision of 0.375 (9/24).

**Reproducibility:**

2: Would be hard pressed to reproduce the results. The contribution depends on data that are simply not available outside the author's institution or consortium; not enough details are provided.

**Reviewer Confidence:**

3: Pretty sure, but there's a chance I missed something. Although I have a good feel for this area in general, I did not carefully check the paper's details, e.g., the math, experimental design, or novelty.

**Typos Grammar Style And Presentation Improvements:**

I think the “contradiction” and “contradictory” are used too liberally where they probably refer to antonym and antonymous in some context; please revisit.
There are many scattered heuristics throughout the methods that also require referring back and forth with the appendix, making it very hard for the reader to track which is which in the end.
English needs editing.

---

> ### Author Rebuttal · Authors · 2023-08-27
>
> Thank you for your reviews and careful read of our paper, we hope to address your comments below!
>
> > Many of the proposed heuristics are crude (e.g., the example at line 940 does not clearly differentiate the subject “the mutant” vs “the chev mutants”), which cast doubt on the promising results in Table 3 with regard to the validity of the sentence-based evaluation. Determining contradictions in medical research is very context-sensitive, and even minor misalignment of the settings can make a detected pair useless. A more realistic test scenario is probably what is reported in section 5.3.3, with a precision of 0.375 (9/24).
>
> Thank you for your points, we believe that they are valid concerns. Our research falls under the umbrella of distant supervision where noisier sources of supervision are used to create larger training sets quickly. Although they are noisy, when coupled during fine-tuning the results are promising, as seen in Table 3. Similar results of distant supervision using noisier data have produced similar trends in other fields (Purver and Battersby, 2012; Zeng et al., 2015).
> Although the experiment in 5.3.3 is an interesting one, in this work we focus on sentence-level contradiction and the main benchmarks in the medical NLI field. We include Section 5.3.3 to show the merit on broader tasks such as document contradiction. We are hopeful that with the success on the sentence-level which we presented in our paper, we will be able to more rigorously define and extend to the document-level task through additional research.
>
> > Is the label at Algorithm 1 Line 13 manually assigned? If automated, how?
>
> The label at Algorithm 1 Line 13 is not manually assigned. The 2 phrases are composed of their respective words, but we ignore the words which are present in both phrases (i.e. we focus on the unique words). The label is assigned based on whether the unique words are synonyms. This check is done via WordNet, since most of the unique words are non-medical parts of speech like adjectives.
>
> > Is Mintz et al’s 2009 work a valid comparison in the context of section 4.2.2? Please justify.
>
> We believe that Mintz et al.’s 2009 work is a valid comparison in the context of Section 4.2.2. In Section 7.2 of Mintz et al.’s work they explain that they manually evaluate 100 samples for a given relation. Similarly, we manually evaluate 100 samples for our contradiction relation. We also follow their evaluation criteria, by assigning “truth” or “falsehood” to our contradiction relationship.
>
> > Are you able to publicly share the raw annotations based on which the 91% and 82% accuracies are claimed in 4.2.1 and 4.2.2?
>
> Yes, we will share those publicly with the rest of the data. Some example are included in Appendix D, and here are a couple more:
>
> *Correctly Labeled Contradiction Example*
>
> **“46 patients (66%) had difficulty talking and 36 (51%) reported difficulty eating.”**
>
> ---
>
> **“only 4 (8%) of 49 patients reported mouth pain and all patients reported being able to eat well.”**
>
> *Correctly Labeled Non-Contradiction Example*
>
> **“we report the third case of persistent polyclonal b-cell lymphocytosis in a male.”**
>
> ---
>
> **“he manifested a lymphocytic leukemoid reaction characterized by marked lymphocytosis and the presence of lymphoblasts in his circulation.”**
>
> > Were the section 3.1.3’s 149 manually verified pairs the only ones used in training the NLI models? What is the difference between the 79% accuracy in 3.1.3 versus the 82% accuracy in in 4.2.2?
>
> The manually verified pairs in Section 3.1.3 are SNOMED phrase terms, where each phrase can be thought of as a node in the SNOMED graph. The 79% accuracy is a reflection of Algorithm 1’s performance.
> Section 4.2.2 on the other hand discusses the samples from our SNOMED dataset, where each instance is a pair of naturally-occurring sentences from PubMed. The 82% accuracy reported is a reflection of the quality of the SNOMED dataset.
> The 149 manually verified phrases from Section 3.1.3 were not used in training any of the NLI models, but rather used in the creation of the SNOMED dataset. The SNOMED dataset was in turn used for fine-tuning the NLI models.
>
>
> > Reproducibility, Typos, Grammar, Style, And Presentation Improvements
>
>  The code is shared via the link at the bottom of Page 2. We realize that we were not able to upload all of the data due to the zip file size limit in the submission, we will make all data publicly available. Thank you for the input on improving the readability of the paper, catching typos, and other presentation issues, we will make sure to fix them in the next iteration of the paper.

---

### Official Review · Reviewer_Kpwr · 2023-08-04

**Soundness:** 4

**Excitement:**

4: Strong: This paper deepens the understanding of some phenomenon or lowers the barriers to an existing research direction.

**Missing References:**

N/A

**Paper Topic And Main Contributions:**

- This paper introduces an approach of using publicly available clinical ontology towards creating a dataset of contradictions using sentences taken from medical abstracts.
- In particular, this paper employs a distant supervision method that utilizes attributes of clinical terminology defined in SNOMED to create pairs of contradicting terms across different medical domains
- Supervised fine-tuning of a number of transformer models (derivatives of BERT and GPT) augmented by the dataset created improves the SOTA performance on a number of labeled datasets for clinical contradiction.


**Questions For The Authors:**

- In Section, 3.1.3 and in Appendix A.1, it is specified that 149 phrase pairs from SNOMED were labeled by annotators? How were these 149 phrase pairs selected? And is there any justification for this specific number?
- In Section 4.1.2, it is mentioned that negation words were removed. How was this performed?
- In Section 4.2.2, it is mentioned that 100 instances were sampled. Can more details be provided about the sampling? Also, how many instances of each label were present in this sample set?
- Can this method of distant supervision be applied towards building datasets for other healthcare-related NLP tasks (beyond contradiction)?

**Reasons To Accept:**

- The paper presents an innovative and relatively inexpensive method of data augmentation for detection of clinical contradiction by leveraging the existing SNOMED medical database

- The paper is well-explained and evidence is provided for most assumptions and modeling decisions.

**Reasons To Reject:**

No reasons to reject but I do have a few questions on the analysis (as specified in the "Questions For The Authors" section)

**Reproducibility:**

3: Could reproduce the results with some difficulty. The settings of parameters are underspecified or subjectively determined; the training/evaluation data are not widely available.

**Reviewer Confidence:**

4: Quite sure. I tried to check the important points carefully. It's unlikely, though conceivable, that I missed something that should affect my ratings.

**Typos Grammar Style And Presentation Improvements:**

- Under section 2, "Related Work", the term RTE is not defined / introduced.

---

> ### Author Rebuttal · Authors · 2023-08-27
>
> Thank you for your insightful questions, we hope to address them below!
>
> > In Section, 3.1.3 and in Appendix A.1, it is specified that 149 phrase pairs from SNOMED were labeled by annotators? How were these 149 phrase pairs selected? And is there any justification for this specific number?
>
> As can be seen in Figure 2 and stated on Line 578, we set the group size to be 25 with 10 samples. To narrow down the phrase pairs, we focused on cardiology related phrase pairs, yielding 750 pairs. We decided to label 20% of these pairs, while filtering phrases which had numbers as part of their phrase. This left us with 149 phrase pairs to be labeled.
>
> > In Section 4.1.2, it is mentioned that negation words were removed. How was this performed?
>
> This was performed manually by our annotators. The objective was to rephrase the sentence as to keep semantic meaning, while not using negation words.
>
> > In Section 4.2.2, it is mentioned that 100 instances were sampled. Can more details be provided about the sampling? Also, how many instances of each label were present in this sample set?
>
> The instances were sampled entirely at random. 32 instances were non-contradictory and 68 were contradictory. We will make sure to add this information in the paper.
>
> > Can this method of distant supervision be applied towards building datasets for other healthcare-related NLP tasks (beyond contradiction)?
>
> One potential application of distant supervision could be towards creating a medical NER dataset. Many of the terms within SNOMED are named entities which can be grouped by their parents’ roots for example. This would allow for a faster way to create NER datasets. In addition, since the nodes contain many attributes - such as synonyms or alternate labels - these could be used to create even larger datasets. For example the sentence “The patient had a heart attack” would have the word ‘heart attack’ distantly labeled as a named entity. Simultaneously, the sentence “The patient had a myocardial infarction” could be synthetically created through substituting the named entity with its synonym. We will add this in the conclusion when discussing future work.
>
> > Reproducibility, Typos, Grammar, Style, And Presentation Improvements
>
>  The code is shared via the link at the bottom of Page 2. We realize that we were not able to upload all of the data due to the zip file size limit in the submission, we will make all data publicly available. Thank you for catching all of the typos and other presentation issues, we will make sure to fix them in the next iteration of the paper.

---

### Official Review · Reviewer_ufkc · 2023-08-11

**Soundness:** 5

**Excitement:**

4: Strong: This paper deepens the understanding of some phenomenon or lowers the barriers to an existing research direction.

**Missing References:**

None.

**Paper Topic And Main Contributions:**

This article introduces a dataset and weakly supervised method -- using domain knowledge through SNOMED -- for contradiction detection from naturally occurring statements in biomedical literature. Extensive evaluations showing SotA results are reported, including formative evaluations to justify design choices, dataset analyses, meaningful comparison with SotA methods and benchmark datasets, ablation studies, and broader evaluations. Overall, the writing is clear, the topic is well-aligned, and the methodology and experimentation sound.

**Questions For The Authors:**

* L373: Why not use medical semantic textual similarity methods instead of one-hot vectors with cosine?
* L408: What is the added scientific value of the Hard-Cardio dataset? It seems some portions of the Appendix might be more valuable for the space taken.
* Why not also evaluate with fine-tuning on only naturally-occurring samples (i.e. the SNOMED dataset)?
* Why not annotate and reserve enough of the SNOMED dataset to do testing on it?
* L559: What exactly does "samples averaged across the models" mean?
* L596: Why not use medical semantic textual similarity methods instead of the PubMed Related Articles metric?
* L599: simalrity -> similarity

**Reasons To Accept:**

This paper adds depth to the discussion on NLI by thoroughly considering the subtask of medical contradiction detection. It provides SotA results and has exemplary writing, meaningful comparison with established literature, and excellent evaluation.

**Reasons To Reject:**

None.

**Reproducibility:**

5: Could easily reproduce the results.

**Reviewer Confidence:**

4: Quite sure. I tried to check the important points carefully. It's unlikely, though conceivable, that I missed something that should affect my ratings.

**Typos Grammar Style And Presentation Improvements:**

* Title: I would argue that this should be considered "medical contradiction detection" rather than "clinical" because, although it uses SNOMED (which has clinical care in view) for the weak supervision, the evaluation is done on biomedical literature. Statements in biomedical literature are general statements about a disease or treatment; statements in clinical text are statements about a patient. These sociolinguistic settings differ significantly. "Medical" was also used in MedNLI and is sufficiently broad to cover both types of texts/settings.
* L058: "Consider the highly domain-specialized example below" rather than talking about ACE inhibitors
* L521: Give the legend for the asterisk in the caption or just below Table 3.

---

> ### Author Rebuttal · Authors · 2023-08-27
>
> Thank you for your review and your meaningful feedback!
> > L373: Why not use medical semantic textual similarity methods instead of one-hot vectors with cosine?
>
> We did some initial testing with medical semantic textual similarity, but empirically did not find a significant difference with the more naive and quickly computable one-hot vector cosine methodology. We will add this experiment to the appendix.
>
> > L408: What is the added scientific value of the Hard-Cardio dataset? It seems some portions of the Appendix might be more valuable for the space taken.
>
> The added scientific value of the Hard-Cardio dataset is that it is a semantically equivalent dataset to Cardio, but we empirically found that models perform worse on it due to the removal of negation phrases. We address this in Section 4.1.2, but will emphasize the added scientific value.
>
> > Why not also evaluate with fine-tuning on only naturally-occurring samples (i.e. the SNOMED dataset)?
>
> We are interested in showing the added value of fine-tuning on the SNOMED dataset. We thought that the best way to show this would be through reporting the relative improvement via additional fine-tuning on the dataset, as fine-tuning on the training set is a common practice reported in baseline papers (Alamri and Stevenson, 2016; Romanov and Shivade, 2018).
>
> > Why not annotate and reserve enough of the SNOMED dataset to do testing on it?
>
> The primary reason for this was because we wanted to evaluate against existing datasets instead of introducing a new evaluation benchmark. Our primary goal in the creation of the SNOMED dataset was for training/fine-tuning as opposed to evaluation.
>
> > L559: What exactly does "samples averaged across the models" mean?
>
> The wording here is a bit confusing. In Figure 2, there are lines for 10, 25, and 50 (green, blue, and red respectively) samples for a given group size. For instance, the blue solid line represents the average ROC AUC of all the small models. The blue dashed line represents the average ROC AUC of all the large models. Small models are defined to be those under 30 million parameters (L543-L546). We will clarify it in the final version.
>
> > L596: Why not use medical semantic textual similarity methods instead of the PubMed Related Articles metric?
>
> Since this was an extrinsic experiment, we decided to proceed with what was established as working well (Castro et al., 2015). This would be an interesting exploration for future work. It would be also interesting to see whether semantic textual similarity methods degrade as context increases.
>
> > All Typos, Grammar, Style And Presentation Improvements
>
> The point about the title is a great one and we will consider it strongly. Thank you for catching all of the typos and other presentation issues, we will make sure to fix them in the next iteration of the paper.

---

### Meta-Review · Area_Chair_xqN6 · 2023-09-04

**Recommendation:** 4
**Confidence:** 3

**Metareview:**

Quality:
Pros with regards to quality are noted by two of the reviewers as follows:
-	“This paper adds depth to the discussion on NLI by thoroughly considering the subtask of medical contradiction detection. It provides SotA results and has exemplary writing, meaningful comparison with established literature, and excellent evaluation.”
-	“The paper is well-explained and evidence is provided for most assumptions and modeling decisions.”

Cons with regards to quality are noted by one reviewer as follows:
-	“Many of the proposed heuristics are crude (…),which cast doubt on the promising results in Table 3 with regard to the validity of the sentence-based evaluation. Determining contradictions in medical research is very context-sensitive, and even minor misalignment of the settings can make a detected pair useless. A more realistic test scenario is probably what is reported in section 5.3.3, with a precision of 0.375 (9/24).”
The authors rebut this point by stating “Our research falls under the umbrella of distant supervision where noisier sources of supervision are used to create larger training sets quickly. Although they are noisy, when coupled during fine-tuning the results are promising, as seen in Table 3.”
The reviewer further notes the mention of WordNet in the rebuttal but not in the paper, which exacerbates concerns around reproducibility. The authors note that the full experiments are fully reproducible via the shared code, i.e. “If you navigate to our code repo (bottom of Page 2), in the file "snomed/snomedct_arr.ipynb", in the function "use_synonym_heuristic", you will see the use of WordNet. All details can be found and reproduced via the code.”
I feel that the point about reproducibility has been well addressed by having the code shared. The point about the crude heuristics is a more challenging one, however, the authors do a good job in pointing this out as a limitation of weak supervision and highlight the added value despite the presence of minor misalignments.

Clarity:
The reviewers disagree with regards to the clarity of the paper. Reviewer 1 states that the paper “has exemplary writing, meaningful comparison with established literature, and excellent evaluation”, while Reviewer 3 notes that “There are many scattered heuristics throughout the methods that also require referring back and forth with the appendix, making it very hard for the reader to track which is which in the end. English needs editing.”

Originality and Significance:
-	“The paper presents an innovative and relatively inexpensive method of data augmentation for detection of clinical contradiction by leveraging the existing SNOMED medical database.”
-	“a well-chosen, meaningful challenge in terms of matching what the NLI technical framework could be conveniently adapted to tackle in the medical domain.”

Overall, I believe that the points raised by reviewer 3 have been meaningfully addressed by the authors, with reproducibility further guaranteed through the shared code.

---

### Decision · Program_Chairs · 2023-10-07

**Decision:**

Accept-Main

**Comment:**

Quality:
Pros with regards to quality are noted by two of the reviewers as follows:
-	“This paper adds depth to the discussion on NLI by thoroughly considering the subtask of medical contradiction detection. It provides SotA results and has exemplary writing, meaningful comparison with established literature, and excellent evaluation.”
-	“The paper is well-explained and evidence is provided for most assumptions and modeling decisions.”

Cons with regards to quality are noted by one reviewer as follows:
-	“Many of the proposed heuristics are crude (…),which cast doubt on the promising results in Table 3 with regard to the validity of the sentence-based evaluation. Determining contradictions in medical research is very context-sensitive, and even minor misalignment of the settings can make a detected pair useless. A more realistic test scenario is probably what is reported in section 5.3.3, with a precision of 0.375 (9/24).”
The authors rebut this point by stating “Our research falls under the umbrella of distant supervision where noisier sources of supervision are used to create larger training sets quickly. Although they are noisy, when coupled during fine-tuning the results are promising, as seen in Table 3.”
The reviewer further notes the mention of WordNet in the rebuttal but not in the paper, which exacerbates concerns around reproducibility. The authors note that the full experiments are fully reproducible via the shared code, i.e. “If you navigate to our code repo (bottom of Page 2), in the file "snomed/snomedct_arr.ipynb", in the function "use_synonym_heuristic", you will see the use of WordNet. All details can be found and reproduced via the code.”
I feel that the point about reproducibility has been well addressed by having the code shared. The point about the crude heuristics is a more challenging one, however, the authors do a good job in pointing this out as a limitation of weak supervision and highlight the added value despite the presence of minor misalignments.

Clarity:
The reviewers disagree with regards to the clarity of the paper. Reviewer 1 states that the paper “has exemplary writing, meaningful comparison with established literature, and excellent evaluation”, while Reviewer 3 notes that “There are many scattered heuristics throughout the methods that also require referring back and forth with the appendix, making it very hard for the reader to track which is which in the end. English needs editing.”

Originality and Significance:
-	“The paper presents an innovative and relatively inexpensive method of data augmentation for detection of clinical contradiction by leveraging the existing SNOMED medical database.”
-	“a well-chosen, meaningful challenge in terms of matching what the NLI technical framework could be conveniently adapted to tackle in the medical domain.”

Overall, I believe that the points raised by reviewer 3 have been meaningfully addressed by the authors, with reproducibility further guaranteed through the shared code.